# Ripe Tomato Saponin Esculeoside A and Sapogenol Esculeogenin A Suppress CD4+ T Lymphocyte Activation by Modulation of Th2/Th1/Treg Differentiation

**DOI:** 10.3390/nu14102021

**Published:** 2022-05-11

**Authors:** Jian-Rong Zhou, Rie Yamada, Erina Huruiti, Nozomi Kitahara, Honami Nakamura, Jun Fang, Toshihiro Nohara, Kazumi Yokomizo

**Affiliations:** Faculty of Pharmaceutical Sciences, Sojo University, Kumamoto 860-0082, Japan; g1551122@m.sojo-u.ac.jp (R.Y.); g1551104@m.sojo-u.ac.jp (E.H.); g1151034@m.sojo-u.ac.jp (N.K.); g1151124@m.sojo-u.ac.jp (H.N.); fangjun@ph.sojo-u.ac.jp (J.F.); none@ph.sojo-u.ac.jp (T.N.); yoko0514@ph.sojo-u.ac.jp (K.Y.)

**Keywords:** CD4+ T lymphocyte, esculeogenin A (Esg-A), esculeoside A (EsA), ripe tomato fruit saponin, Th1, Th2, Treg

## Abstract

We report that esculeoside A (EsA), a glycoside and a major component in ripe tomato fruit, ameliorated experimental dermatitis in mice. However, the underlying immunologic molecular mechanisms are unknown. The present study examined its underlying immune nutrition mechanism using concanavalin A (ConA)-blast mouse splenocyte primary culture. We found that EsA and its sapogenol esculeogenin A (Esg-A) concentration-dependently suppressed T-lymphoproliferation using CFSE-labeled flow-cytometry and water-soluble tetrazolium (WST) assay. Using ELISA and *q*-PCR methods, EsA/Esg-A showed profound decreases in T helper 2 (Th2)-relevant interleukin-4 (IL-4) secretion and mRNA expression, and GATA3 expression. Moreover, EsA/Esg-A suppressed CD4+ T-lymphocyte activation by decreasing IL-2 secretion and mRNA expression and CD25+ cell proportion. Further, EsA/Esg-A alleviated Treg suppressive activity by reducing IL-10 secretion, Foxp3 mRNA expression, and cell numbers. We suggest the immune nutrition function by tomato component, and highlight that EsA/Esg-A are capable of reducing CD4+ T-lymphocyte activation via a reduction in Th2-lymphocyte activity by modulation of Th2/Th1/Treg subunit differentiation.

## 1. Introduction

Eesculeoside A (EsA), a spirosolane-type steroidal alkaloid glycoside from fresh tomato fruit (*Lycopersicon esculentum*) [1], is a major component of ripe tomato and is present at a four-fold higher content than that of lycopene, the main tomato carotenoid [2,3]. We previously found that oral treatment with this tomato saponin at 10, 50, and 100 mg/kg bodyweight ameliorated 2,4-dinitrochlorobenzene-induced experimental dermatitis in mice. However, the underlying dominant immune nutrition mechanism by EsA is still unknown. Moreover, Nohara et al. found that the final metabolites of EsA were eliminated as androsterone analogues in the urine of men orally administered ripe cherry tomato fruit [4], and it was proposed that spirostanol and furostanol glycosides EsA are changed into its sapogenol esculeogenin A (Esg-A) in the intestine and then metabolized to pharmacologically active pregnane derivatives by the introduction of a hydroxyl group at C-23.

It is known that atopic dermatitis (AD) is a chronic inflammatory disorder predominantly mediated by CD4+ T helper cells [5,6,7,8]. First, an overexpression of Th2 cell cytokines has been observed in acute and chronic lesions of AD. The Th2 response is characterized by skewed cytokine production including IL-4 and IL-5, and associated downstream GATA binding protein 3 (GATA3), which in turn drive eosinophilic inflammation and Ig class switching to IgE in B cells and subsequent release of inflammatory mediators [9]. Second, it was reported that chronic lesions also express moderate levels of the Th1 cell cytokine IFN-γ; thus, the T cell response present in AD is not only Th2-polarized but may lead to heterogeneous cytokine production involving Th1 cell cytokines and Th1 downstream T-box transcription factor 21 (Tbx21, T-bet) [10]. Third, it was reported that children with AD had higher Th2, naïve regulatory T (Treg), and memory Treg-cell numbers compared to healthy children. Werfel also reported that thymus-derived Treg cells were detected in elevated numbers in the circulation of AD patients [8]. Treg cells can be subdivided in two distinct subsets: naïve and memory Tregs; naive Tregs mostly originate from the thymus, while memory Tregs are induced in the periphery. While higher memory Treg cells represent chronic inflammation, naïve cells have no memory of sensitization to specific antigens [11,12,13]. The major population of CD4+ Treg cells was found to be characterized by high expression of IL-2 receptor alpha chain (CD25) and transcription factor Foxp3 (forkhead box P3), the latter being indispensable for the development and suppressive function of Treg cells.

Thus, the present study investigated EsA- and/or Esg-A-induced immune molecular nutrition effects, especially its relevance to CD4+ T lymphocytes following in vitro stimulation with concanavalin A (ConA).

## 2. Materials and Methods

### 2.1. Preparation of EsA and Esg-A

Fresh ripe cherry tomato fruit (*Lycopersicon esculentum* var. *cerasiforme*) were crushed, added with water, and centrifuged. The supernatant was passed through a Diaion^®^ HP-20 (Mitsubishi Chemical, Tokyo, Japan) and separated by gradient elution with 40% aq. methanol, 60% aq. methanol, and methanol, successively. The 60% eluate provided EsA. EsA was checked with thin-layer chromatography, and the average EsA yield was calculated as about 0.043%. EsA was hydrolyzed with 2 N HCl, and then the reaction mixture was extracted with ethyl acetate. The organic layer was evaporated in vacuum to afford a residue, which was purified by silica gel column chromatography with CHCl_3_-methanol-H_2_O = 9:1:0.1 to give Esg-A [2,14]. The chemical structure of EsA is shown in Figure 1.

### 2.2. Animals and Splenocyte Isolation

The study was submitted to and approved by the Ethics Committee of Sojo University (2019-P-025, 2020-P-025). All experiments were conducted in strict accordance with the Guidelines of the Japanese Pharmacological Society for the Care and Use of Laboratory Animals. Female BALB/c mice, 6~9 weeks old, were obtained from Japan SLC (Hamamatsu, Japan). The animals were housed under conditions of controlled room temperature (24.5–25.0 °C), 60 ± 10% humidity, and a 12/12 h light-dark cycle. Food pellets and water were provided ad libitum. Mice were anaesthetized by exposure to isoflurane; the spleen was dissected out and immediately immersed in phosphate-buffered saline (PBS), then minced and passed through a 70 μm cell strainer. After treatment with red blood cell (RBC) lysis buffer to deplete red blood cells, splenocytes were suspended in Roswell Park Memorial Institute (RPMI) 1640 medium (Wako, Osaka, Japan) supplemented with 10% (*v*/*v*) fetal bovine serum (FBS), 100 U/mL penicillin, and 100 μg/mL streptomycin (Invitrogen, Waltham, MA, USA) at a density of 3 × 10^6^ cells/well in 24-well plates, and cultured under a humidified atmosphere of 5% CO_2_ at 37 °C, before being subjected to various treatments.

### 2.3. Cytotoxicity Assay

Splenocytes were exposed to EsA (10, 30, 100, or 300 μM) or Esg-A (1, 3, 10, or 30 μM) and cultured for 24 h. After harvesting, cells were stained with Annexin V-fluorescein isothiocyanate (FITC) and 7-aminoactinomycin D (7-AAD) (eBioscience, Waltham, MA, USA), and analyzed with a BD AccuriTM Plus Flow Cytometer (BD Biosciences, Franklin Lakes, NJ, USA). Cell viability was calculated by counting Annexin V- and 7-AAD-negative cells. EsA or Esg-A as a stock solution (300 mM or 30 mM) was prepared with dimethyl sulfoxide (DMSO).

### 2.4. ConA Proliferation Assay

Splenocyte proliferation was labeled using 5-(6)-carboxyfluorescein diacetate succinimidyl ester (CFSE) (Sigma-Aldrich, Saint Louis, MO, USA) [15]. Splenocytes (1 × 10^7^ cells/mL) were suspended in 5 μM CFSE (5 mM stock solution in DMSO) in PBS and incubated at 37 °C for 10 min. The labeling process was quenched by adding an equal volume of heat-inactivated FBS. After being washed twice, recounted, adjusted to a density of 5 × 10^5^ cells/mL, and seeded onto 24-well plates, CFSE-labeled cells were treated with EsA or Esg-A and stimulated with 1 μg/mL ConA (Sigma-Aldrich), then cultured for 3 days. After harvesting, cells were pretreated with anti-mouse CD16/32 (clone 93, eBioscience) to block nonantigen-specific binding of immunoglobulins, then incubated at 4 °C for 30 min with anti-mouse CD3e conjugated to phycoerythrin (PE) (145-2C11, eBioscience) or the corresponding isotype antibody Armenian hamster IgG isotype control conjugated to PE (eBio299Arm, eBioscience). CFSE-labeled-cells were suspended in the staining buffer, and proliferation was analyzed using flow cytometry.

Splenocyte proliferation was also analyzed by water-soluble tetrazolium (WST) assay using a Cell Count Kit-8 (CCK-8) (Dojindo, Kumamoto, Japan). After seeding, splenocytes were pretreated with EsA or Esg-A, stimulated with ConA, and cultured for 3 days. Next, the cells were incubated with CCK-8 for 5 h, and the optical density was read at a wavelength of 450 nm using an Infinite^®^ 200 PRO microplate reader (Tecan Group Ltd., Seestrasse, Switzerland). The proliferation rate was compared with that of the untreated control group.

### 2.5. Enzyme-Linked Immunosorbent Assay (ELISA)

Splenocytes were treated with EsA or Esg-A for 1 h before stimulation with ConA (1 μg/mL). After 24 or 48 h, cell-free culture supernatants were collected and stored at −80 °C until use. The secretion of IL-2, IL-4, IL-10, and transforming growth factor-β (TGF-β) were quantified using the corresponding mouse ELISA kits (88-7024 for IL-2, 88-7044 for IL-4, 88-7104 for IL-10, and 88-8350 for TGF-β, eBioscience) according to the manufacturer’s instructions.

### 2.6. Quantitative PCR (RT-qPCR) Assay

Quantification of mRNA expression in splenocytes was performed using RT-qPCR by subjecting the cDNA obtained from the following preparation methods to PCR amplification using a StepOnePlus™ Real-Time PCR System (Life Technologies, Waltham, MA, USA).

Total RNA from mouse splenocytes was isolated from homogenates using a RNEasy^®^ Mini Kit (Qiagen, Düsseldorf, Germany). The obtained mRNA was quantified by measuring the absorbance at 260 nm, and its quality was determined by measuring the 260/280 ratio. cDNA was synthesized using a PrimeScriptTM 1st strand cDNA Synthesis kit (Takara Bio, Kusatsu, Japan). In brief, 1.0 μg of total RNA from each sample was added to a mixture of 1.0 μL of random 6 mers (50 μM), 1.0 μL of dNTP mix (10 mM), and 8.0 μL of RNase-free distilled H_2_O (dH_2_O), and then held at 65 °C for 5 min and cooled to 4 °C using a thermocycler (Eppendorf, Hamburg, Germany). Next, the denatured mixture was added with 4.0 μL of 5× PrimeScript buffer, 0.5 μL of RNase inhibitor, 1.0 μL of PrimeScript RTase, and 4.5 μL of RNase-free dH_2_O, and then held at 50 °C for 45 min, heated to 95 °C for 5 min, and cooled to 4 °C. Finally, the 20 μL PCR reaction mixture contained 1.2 μL of 10 M forward primers and 1.2 μL of 10 M reverse primers (300 nM final concentration of each primer), 10 μL of PowerUpTM SYBR^®^ Green Master mix (2×), 5.6 μL of nuclease-free water, and 2.0 μL of the obtained cDNA. Thermal cycling conditions were applied as follows: 50 °C for 2 min, 95 °C for 2 min, followed by 40 cycles of 15 s at 95 °C and 1 min at 60 °C. The samples were matched to a standard curve generated by amplifying serially diluted products using the same real-time PCR conditions. The data are presented as the fold change (2^−ΔΔCt^) in gene expression levels and normalized to the mRNA expression levels of an endogenous reference gene, glyceraldehyde 3-phosphate dehydrogenase (GAPDH), then shown relative to that of the control group [16].

The primer sequences of the target genes, selected from PubMed and other databases, were as follows (Table 1):

### 2.7. Flow Cytometry Cell Surface and Transcriptional Factor Staining

After treatment with EsA, Esg-A, and ConA for 24 h, splenocytes were harvested and washed twice in PBS. The expressions of CD4, CD25, and Foxp3 were quantified using a Foxp3/transcription factor staining kit (88-8111, eBioscience) according to the manufacturer’s instructions. Briefly, cells were first preincubated with anti-mouse CD16/32 to prevent nonspecific binding of antibodies to FcγR (Fc gamma receptor). Then, the cell surface was stained with the fluorochrome-labeled antibodies anti-mouse CD4 conjugated to FITC (RM 4-5) and anti-mouse CD25 conjugated to APC (PC61.5, allophycocyanin) or the corresponding isotype antibody rat IgG2a conjugated to FITC (eBR2a) and rat IgG1 isotype control conjugated to APC (eBRG1). After incubation, the cells were washed with staining buffer, and fixed with fixation working solution. After wash, the cells were then permeabilized using permeabilization buffer, and the nuclear protein was stained with anti-mouse/rat Foxp3 conjugated to PE (FJK-16S) or the corresponding isotype antibody rat IgG2a isotype control conjugated to PE (eBR2a). After wash with the permeabilization buffer, the cells were then resuspended with 200 μL of staining buffer and filtered for analysis using BD Accuri C6 Plus flow cytometer. 

### 2.8. Statistical Analysis

The data are presented as the mean ± SEM of at least three independent experiments and were analyzed by Prism 8 (GraphPad Software, San Diego, CA, USA). Student’s *t*-test was used to determine statistical significance between two groups, and one-way ANOVA with Tukey’s multiple comparison test was used for multiple groups. *p*-values < 0.05 were considered to be significantly different.

## 3. Results

### 3.1. Cytotoxicity of EsA/Esg-A in Splenocytes

Considering the previous in vitro and in vivo effects [1,17], various concentrations of EsA and Esg-A were tested for cytotoxicity to splenocytes using flow cytometry (Figure 2). When the live cell percentage of the control was normalized to 1.0, the ratios of live splenocytes were larger than 0.9 in the presence of EsA (10 to 300 μM) or Esg-A (1 to 30 μM). Further, the EsA/Esg-A average percentage of apoptotic splenocytes (7-AAD negative and Annexin V-FITC positive) were similar with that of the control.

### 3.2. Suppression of T Lymphocyte Proliferative Response by EsA/Esg-A

Effects of EsA/Esg-A on the T lymphocyte proliferative response were tested using in vitro ConA-stimulated T cell proliferation (Figure 3). The T lymphocyte proliferative effect in the culture with EsA/Esg-A was compared to that without EsA/Esg-A (ConA alone). As shown in Figure 3D using the WST assay, addition of EsA/Esg-A to the culture system resulted in a concentration-dependent suppression of the proliferation response. Esg-A at about 8 μM inhibited the elevated proliferation ratio by 50%, whereas EsA at about 100 μM was required to show the same inhibition. In Figure 3A–C, further assays using CD3-PE+/CFSE+ fluorescence demonstrated whether this dose-dependent suppression by EsA/Esg-A was related to T cell division. Daughter T cells (Figure 3A, R2), derived from responder splenocytes following ConA stimulation, were differentiated from undivided T cells (R1) and identified by the intensity of CFSE staining (Figure 3A,B) using flow cytometry. The addition of EsA/Esg-A to CFSE-labeled cells stimulated with ConA showed a tendency to suppress T lymphocyte division (Figure 3B,C). Esg-A at about 10 μM decreased dividing T cells by 50%, whereas about 300 μM EsA was needed to show the same reduction. 

### 3.3. EsA/Esg-A Modulation of ConA Blast-Induced Cytokine Production

During T cell activation, cytokines are produced to modulate the differentiation and subsequent specialization of T cells; thus, we examined the effect of EsA/Esg-A on ConA blast-induced cytokine extracellular secretion. EsA/Esg-A decreased the production of IL-2 and the Th2 cytokine IL-4 in a concentration-dependent manner, as shown in Figure 4A. In detail, Esg-A at about 10 μM and 5 μM inhibited the elevated IL-2 and IL-4 production by 50%, whereas EsA at about 150 μM and 100 μM was required to show the same inhibition, respectively, indicating that the inhibitory effect of EsA/Esg-A on IL-4 cytokine production is greater than for IL-2 production. Moreover, EsA/Esg-A also decreased production of the Treg cytokine IL-10, but did not affect that of TGF-β, and both are required for Treg cell maintenance [18]. We also investigated their responses for 48 or 72 h, and there were similar effects as those for 24 h (data not shown).

### 3.4. EsA/Esg-A Modulation of ConA Blast-Induced Cytokine Gene Expression

We also examined the effect of EsA/Esg-A on ConA blast-stimulated cytokine mRNA expression. EsA/Esg-A concentration-dependently decreased mRNA expression levels of IL-2, IL-4, and IFN-γ, as shown in Figure 4B. In detail, Esg-A at about 30 μM, 20 μM, and 20 μM inhibited the elevated IL-2, IL-4, and IFN-γ gene expression levels by 50%, respectively, whereas for EsA > 100 μM, about 60 μM and 100 μM were required to show the same inhibition, indicating that the inhibitory effect of EsA/Esg-A on IL-4 gene expression level is greater than those on IL-2 and IFN-γ expression. However, EsA/Esg-A did not affect the TGF-β gene expression level.

### 3.5. EsA/Esg-A Suppression of Th2/Th1/Treg Master Gene Expression

We then examined the effects of EsA/Esg-A on ConA blast-stimulated transcription factor expression of Th2/Th1/Treg. As shown in Figure 5, EsA/Esg-A decreased mRNA expression levels of GATA3, Tbx21, and Foxp3. In detail, Esg-A at about 10 μM, 20 μM and 30 μM inhibited the elevated GATA3, Tbx21 and Foxp3 expression levels by 50%, whereas EsA at about 20 μM, 100 μM and >100 μM was required to show the same inhibition, respectively. Thus, the inhibitory effects of EsA/Esg-A on Th2 master gene (GATA3) expression level are greater than those on Th1 master gene (Tbx21) and Treg master gene (Foxp3) expression.

### 3.6. EsA/Esg-A Suppression of T Lymphocyte Activation and Treg Cell Proportion

We finally examined the effects of EsA/Esg-A on ConA blast-stimulated T lympho- cyte activation and Treg cell proportion. As shown in Figure 6, ConA stimulation did not affect CD4+ T cell proportion, but significantly increased CD25+/CD4+ cell proportion (a marker of T cell activation) from 4.6 to 58.6%, and also Foxp3+/CD25+/CD4+ cell proportion from 3.7 to 14.8%. The addition of EsA/Esg-A decreased CD25+/CD4+ and Foxp3/CD25+/CD4+ T cell proportion in a concentration-dependent manner. Specifically, about 20 μM Esg-A inhibited CD25+/CD4+ T cell proportion by 50%, whereas about 100 μM EsA was required to show the same inhibition. Moreover, Foxp3/CD25+/CD4+ T cell proportions (%) were 14.36, 13.1, and 10.38 in the presence of EsA at 10, 30, and 100 μM, and were 12.12, 11.54, and 10.44 in the presence of Esg-A at 1, 3, and 10 μM, respectively. Thus, the EsA/Esg-A-induced inhibition ratio for the CD25+/CD4+ T cell proportion is greater than that for the Foxp3/CD25+/CD4+ T cell proportion.

## 4. Discussion

The present study provides the first evidence that the saponin EsA and its sapogenol Esg-A from ripe tomato fruit alleviate ConA-blast T lymphocyte activity by modulation of Th1/Th2/Treg-associated cytokines and transcription factor signaling.

Balb/c female mouse is preferred for immunology studies including T cell polarization, interleukin profiling, and allergenicity, and it was reported that female mice develop a more pronounced type of allergic inflammation than male mice [19]. We also reported that esculeoside B (EsB), a solanocapsine-type glycoside and a major component in tomato juice, ameliorated experimental dermatitis in Balb/c female mice through decreases in IgE and ConA-mitogenic action, and a decline in IL-4 secretion [20]. Hence, the present study investigated the effects of EsA/Esg-A on ConA-blast Balb/c female mouse splenocytes, and analyzed the immune nutrition molecular mechanism in relation to CD4+ T lymphocytes. We pretested the EsA/Esg-A cytotoxic effects in mouse primary splenocytes, and found that apoptosis was lower at EsA < 300 μM and Esg-A < 30 μM. Based on these concentrations, we first tested the effects of EsA/Esg-A on T lymphoproliferative action induced by ConA, a selective T cell mitogen. Using flow cytometry, EsA/Esg-A suppressed CD3+ T lymphocyte division and showed profound suppression of splenocyte proliferation by the WST assay. Both assays showed that EsA/Esg-A decreased T lymphocyte proliferation; however, the latter may suggest their possible effect on B lymphocytes, because murine splenocytes mainly contain about 30% T and 50% B lymphocytes. The present T lymphoproliferative decline by EsA is in line with our previous in vivo study using EsB [20].

Second, we tested EsA/Esg-A for their Th2/Th1 cytokine production modulatory potential, since in the atopy patch test, it was observed that T cells in the skin display an initial Th2 polarization, with increasing populations of Th1 cells in patients with chronic AD [21,22,23]. The high proportion of Th2-polarized T cells appears to be a key factor in patients with allergic inflammation [24]. The present study used the ConA-stimulated lymphoproliferative system, and it was found that EsA/Esg-A suppressed both Th2/Th1 cytokine production and intracellular gene expression levels. We subsequently tested EsA/Esg-A for their Th2/Th1 master gene modulatory potential, and found that EsA/Esg-A suppressed both GATA3 and Tbx21 expression levels associated with Th2- and Th1-specific genes, respectively. Moreover, the present study also indicated that the EsA/Esg-A inhibitory effects on Th2-mediated cytokine production and cytokine gene and master gene expression were greater than those on Th1-mediated responses. Recently, the clinical efficacy of the IL-4 receptor antagonist dupilumab was demonstrated, in addition to a decline in Th1 and Th2 cell numbers [25,26], supporting evidence for the importance of the Th2/Th1 immune pathway in AD. Thus, we highlight that EsA/Esg-A can suppress Th2/Th1 dominant inflammatory reaction.

Third, in addition to Th2 and Th1 effector cells, CD4+ Th cells are controlled by Treg cells, which regulate inflammatory responses and restore immune homeostasis. We examined the Treg modulatory potential of EsA/Esg-A, and demonstrated that EsA/Esg-A suppressed the secretion and gene expression of Treg cytokine IL-10, and decreased transcription factor Foxp3 expression and Foxp3+/CD25+/CD4+ T cell proportion. This indicated that EsA/Esg-A alleviated the suppressive action of IL-10-producing Treg cells in mouse splenocytes in response to ConA mitogenic stimuli. At the same time, EsA/Esg-A modulation in CD4+ Th cells is characterized by a reduction in IL-2 signaling including IL-2 receptor alpha chain (CD25) and IL-2 cytokine gene expression and secretion. However, the expression of CD25, as a marker of T cell activation, is tightly restricted to peripheral-activated T, B, and Treg cells [27]. Figure 6 shows the profound suppression of CD25+CD4+ T cells, further suggesting that EsA/Esg-A inhibit the activation of CD4+ T lymphocytes, and then regulate CD4+ T cell proliferation and differentiation. At the same time, IL-2 has been reported as an important activator of Treg suppressive activity in vitro and in vivo [28,29,30,31], and the present data suggest that during suppressed IL-2 signaling by EsA/Esg-A, the concomitant suppression of Treg and/or the other immune cells may still counteract ConA-induced in vitro activation of cellular immunity. On the other hand, EsA/Esg-A did not affect TGF-β cytokine secretion and gene expression. That TGF-β signaling limits immune activation in the mouse model of atopic dermatitis is well-established [32,33,34], and the present data suggest that EsA/Esg-A may not suppress TGF-β-stimulated Treg signaling, a point that needs to be examined further. It has been also reported that ConA can stimulate mouse T-cell subsets, giving rise to several functionally distinct T cell populations, including precursors to regulatory T cells [35]. Thus, the present findings might underlie at least partly the mechanism of EsA-amelioration of experimental dermatitis.

Finally, in a comparison of EsA- and Esg-A-suppressive potential in Th2/Th1/Treg activation, the suppression by Esg-A was greater than that by EsA, suggesting that the steroidal alkaloid Esg-A moiety may be mainly responsible for the suppressive effect on CD4+ T cell activity at least in vitro. The main effect of Esg-A should be consistent with our previous report, in which tomato saponin EsA was metabolized into Esg-A in the intestine and then to pharmacologically active pregnane derivatives.

## 5. Conclusions

These novel results clarify the molecular nutrition modulation on acquired immunity cells by tomato saponin EsA and sapogenol Esg-A, which are major components of dietary fresh tomato fruits. Next steps include investigation of their downstream stimuli and the signaling pathways affecting the Th2/Th1/Treg cells, and it is also important to study EsA/Esg-A effects on human PBMCs (peripheral blood mononuclear cells).

## Figures and Tables

**Figure 1 nutrients-14-02021-f001:**
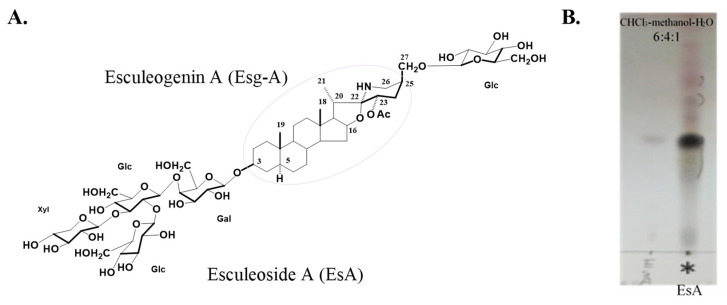
Chemical structures of EsA/Esg-A (**A**). Glc: glucose; Gal: galactose; and Xyl: xylose. Molecular weight: 1270.38 g/mol (EsA); 447.66 g/mol (Esg-A). Thin-layer chromatography of EsA (**B**).

**Figure 2 nutrients-14-02021-f002:**
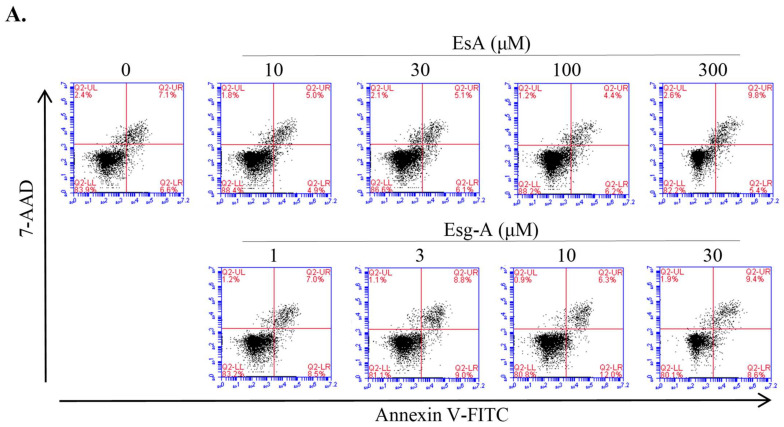
Cytotoxicity of EsA/Esg-A in splenocytes. (**A**) Splenocytes (3 × 10^6^ cells/well) were seeded onto 24-well plates and treated with 0.1% DMSO, or the indicated concentrations of EsA (10, 30, 100, and 300 μM) or Esg-A (1, 3, 10, and 30 μM) for 48 h. Cells were then stained with Annexin V-FITC and 7-AAD, and analyzed by flow cytometry. (**B**) The results represent four independent experiments, and each bar is expressed as mean ± SEM.

**Figure 3 nutrients-14-02021-f003:**
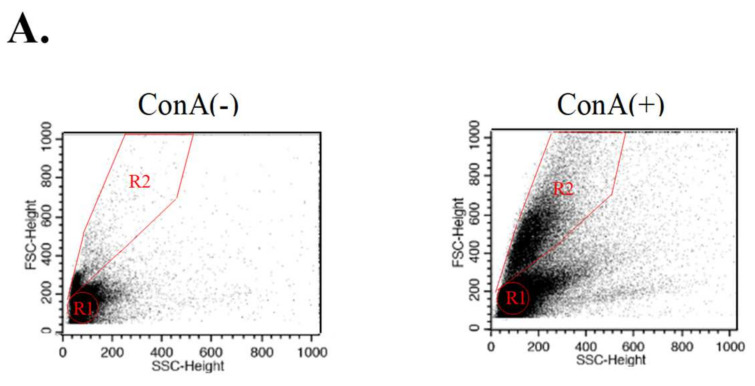
Inhibition of ConA blast proliferation by EsA/Esg-A. (**A**) Cell division in CFSE-labeled splenocytes was determined by flow cytometric analysis. Splenocytes were labeled with CFSE and treated with 0.1% DMSO, or the respective concentrations of EsA/Esg-A for 1 h, then stimulated with ConA (1 μg/mL) for 72 h. Alterations in light scatter characteristics: splenocytes alone (**left**), ConA blast of splenocytes (**right**). All plots were gated on CD3-positive cells including both resting lymphocytes (R1) and blasts (R2). (**B**) Representative histogram plots show the cell division associated with CFSE-labeled splenocytes without and with EsA/Esg-A, while the histograms were gated to both R1 and R2. (**C**) Expression of cell division percentage (M1) by ConA blast without and with EsA/Esg-A addition. Data represent three independent experiments. (**D**) Expression of cell proliferation ratio of ConA blasts without and with EsA/Esg-A addition was determined by the WST assay. Splenocytes were treated with 0.1% DMSO or the respective concentrations of EsA/Esg-A for 1 h, then stimulated with ConA (1 μg/mL) for 48 h. Results were normalized to the optical density of the culture by ConA alone. Data represent five independent experiments, and each bar is expressed as mean ± SEM. *: *p* < 0.05, **: *p* < 0.01, ***: *p* < 0.001, significantly different from the control (ConA alone). -: without ConA treatment; EsA: esculeoside A; Esg-A: esculeogenin A. The grey dotted line shows 50% of ConA-induced elevated responses.

**Figure 4 nutrients-14-02021-f004:**
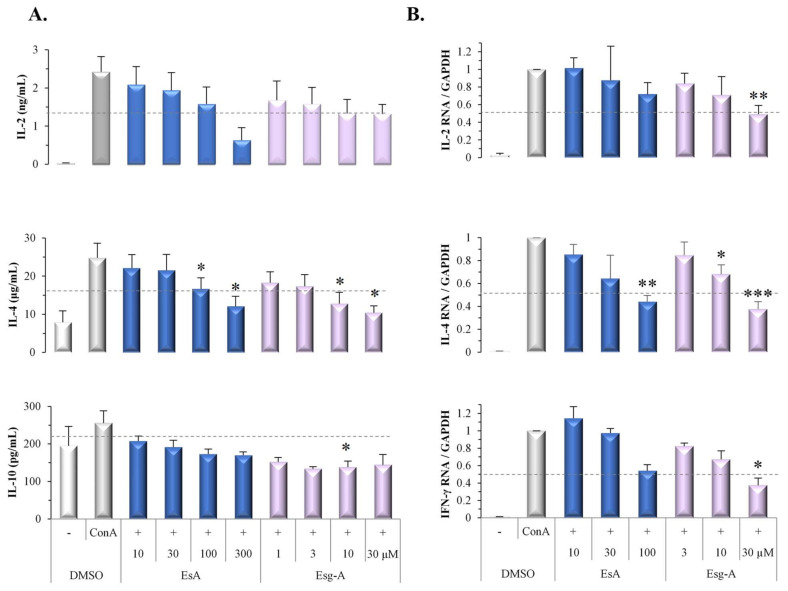
Modulation of ConA blast-induced cytokine production and gene expression by EsA/Esg-A. Splenocytes (3 × 10^6^ cells/well) were pretreated with 0.1% DMSO or the respective concentrations of EsA/Esg-A for 1 h, and then stimulated with ConA (1 μg/mL) for 24 h. The culture supernatants were collected and assayed for cytokine secretion using ELISA. The cells were harvested, and RNA was extracted and reverse transcribed to cDNA. Cytokine mRNA expression was determined by RT-PCR. (**A**) EsA/Esg-A decreased IL-2, IL-4, and IL-10, but not TGF-β production in ConA-stimulated splenocytes. Data represent five independent experiments. (**B**) EsA/Esg-A decreased IL-2, IL-4, and IFN-γ but not TGF-β mRNA expression in ConA-stimulated splenocytes. The relative mRNA expression was normalized to the endogenous control gene GAPDH and calibrated using EsA-Esg-A-untreated (ConA alone) cells. Data represent three independent experiments. Each bar is expressed as mean ± SEM. *: *p* < 0.05, **: *p* < 0.01, ***: *p* < 0.001, significantly different from the control (ConA alone). -: without ConA treatment; EsA: esculeoside A; Esg-A: esculeogenin A. The grey dotted line shows 50% of ConA-induced elevated responses.

**Figure 5 nutrients-14-02021-f005:**
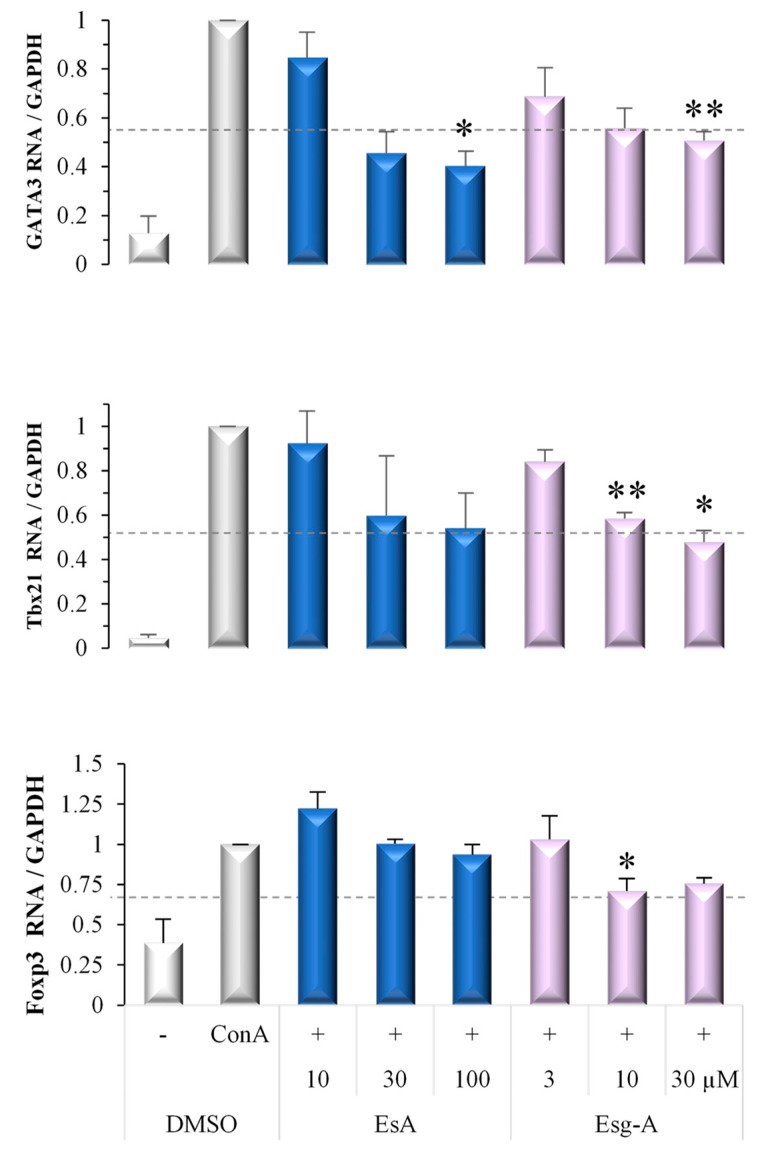
Suppression of ConA blast-induced Th1/Th2/Treg master gene expression by EsA/Esg-A. Splenocytes (3 × 10^6^ cells/well) were pretreated with 0.1% DMSO or the respective concentrations of EsA/Esg-A for 1 h, and then stimulated with ConA (1 μg/mL) for 24 h. The cells were harvested, and RNA was extracted and reverse transcribed to cDNA. Th1/Th2/Treg master gene expression was determined by RT-PCR. EsA/Esg-A decreased GATA3, Tbx21, and Foxp3 mRNA expression in ConA-stimulated splenocytes. The relative mRNA expression was normalized to the endogenous control gene GAPDH and calibrated using EsA-Esg-A-untreated (ConA alone) cells. Data represent three independent experiments. Each bar is expressed as mean ± SEM. *: *p* < 0.05, **: *p* < 0.01, significantly different from the control (ConA alone). -: without ConA treatment; EsA: esculeoside A; Esg-A: esculeogenin A. The grey dotted line shows 50% of ConA-stimulated elevated mRNA expression level.

**Figure 6 nutrients-14-02021-f006:**
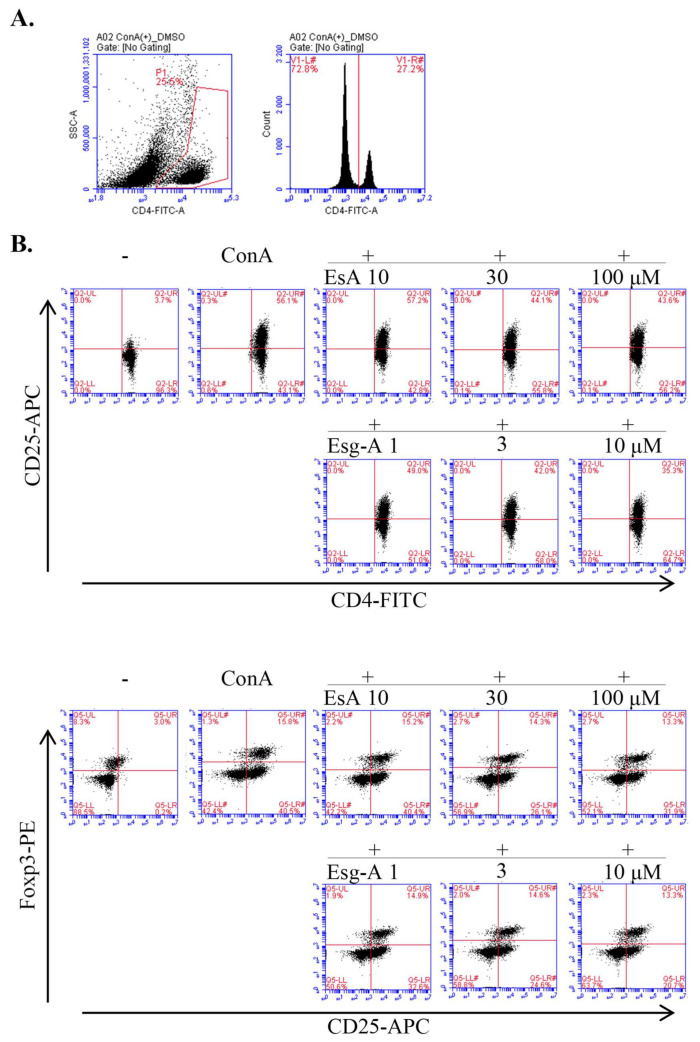
Suppression of CD25^+^ and Foxp3^+^ cell rate by EsA/Esg-A. Splenocytes (3 × 10^6^ cells/well) were pretreated with 0.1% DMSO or the respective concentrations of EsA/Esg-A for 1 h, and then stimulated with ConA (1 μg/mL) for 24 h. The cells were harvested, and the CD4+ Th cell subset was subsequently detected by surface and nuclear staining, and analyzed by three-color flow cytometry. (**A**) Splenocytes were gated on CD4+ Th cells (**left**). The histogram shows the rate of CD4+ Th cells (**right**). (**B**) The representative dot plots show the CD25+ and Foxp3+ percentage in the absence and presence of EsA/Esg-A. (**C**) Effects of EsA/Esg-A on CD4+, CD25+, and Foxp3+ cell percentage. Data represent five independent experiments. Each bar is expressed as mean ± SEM. *: *p* < 0.05, **: *p* < 0.01, significantly different from the control (ConA alone). -: without ConA treatment; EsA: esculeoside A; Esg-A: esculeogenin A. The grey dotted line shows 50% of ConA-stimulated elevated cell proportion.

**Table 1 nutrients-14-02021-t001:** RT-qPCR primers for GAPDH, IL-4, GATA3, IL-2, Tbx21, IFN-γ, TGF-β1, IL-10, and Foxp3 mRNA.

	Forward	Reverse
GAPDH	5′-CCCAGCAAGGACACTGAGCAAG-3′	5′-GGTCTGGGATGGAAATTGTGAGGG-3′
IL-4	5′-GAAGCCCTACAGACGAGCTCA-3′	5′-ACAGGAGAAGGGACGCCAT-3′
Gata3	5′-GGATGTAAGTCGAGGCCCAAG-3′	5′-ATTGCAAAGGTAGTGCCCGGTA-3′
IL-2	5′-TCCAGAACATGCCGCAGAG-3′	5′-CCTGAGCAGGATGGAGAATTACA-3′
Tbx21	5′-CTGCCTACCAGAACGCAGA-3′	5′-AAACGGCTGGGAACAGGA-3′
IFN-γ	5′TCTGGGCTTCTCCTCCTGCGG-3′	5′GGCGCTGGACCTGTGGGTTG-3′
TGF-β1	5′TACGGCAGTGGCTGAACCAA-3′	5′CGGTTCATGTCATGGATGGTG-3′
IL-10	5′GCCAGAGCCACATGCTCCTA-3′	5′GATAAGGCTTGGCAACCCAAGTAA-3′
Foxp3	5′TGCCTTCAGACGAGACTTGGA-3′	5′GGCATTGGGTTCTTGTCAGAG-3′

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
