# Peer review of "Ripe Tomato Saponin Esculeoside A and Sapogenol Esculeogenin A Suppress CD4+ T Lymphocyte Activation by Modulation of Th2/Th1/Treg Differentiation"

_nutrients, 2022, doi:10.3390/nu14102021_

Round 1

Reviewer 1 Report

Undoubtedly, atopic dermatitis in children is an important topic in nutrition. The objective of the study was to determine the modulation of TH1/TH2/Treg immune response by EsA- and/or Esg-A- in splenocytes stimulated with ConA. The main concern is about the use of ConA as stimulus on splenocytes, mainly if the authors consider that this response could be representative of atopic dermatitis.

I have some relevant observations about the model used by the authors:

  1. ¿Do the authors consider the ConA stimulus appropriate to study atopic dermatitis? Could you explain why you didn't use a more appropriate allergen?
  2. ¿What is the reason for using only Balb/c females in the study?
  3. ¿Were the mice exposed to any allergens or inducers of the atopic response?
  4. The time for cytotoxic response is adequate. However, I think 24 hours is inadequate for establishing a TH1 or TH2 immune response, especially if the mice were not previously in contact with an allergen. The authors need to explain why the response of naïve splenocytes stimulated with ConA would be characteristic of an individual with an atopic dermatitis profile?
  5. ¿Were cell death or apoptosis controls used in your flow cytometry studies or splenocyte cultures? Please include them in the materials and methods sections and in the Results.
  6. Figures 3 and 5 could not be fully viewed in its pdf.
  7. The authors mention that Esg-A can be excreted as androgen metabolites ¿could the effects be due to an inhibition mediated by its analogy to glucocorticoids? ¿have you detected levels of lipocortin or phospholipase A2?
  8. In the Discussion section. The first part of the discussion is redundant with the introduction and results section. The authors mention that EsA/Esg-A suppressed both Th2/Th1 cytokine production and ¿intracellular? gene expression levels, however, IFN-ϒ production was not analyzed (higher concentrations of Esg-A are missing in Figure 3).

Author Response

Dear Reviewer,

MDPI Nutrients Editorial Office

We address the following points, and have submitted a redline version of our manuscript.

We sincerely thank your pertinent advices in improving our manuscript.

Kind regards,

Jian-Rong Zhou

Department of Presymptomatic Medical Pharmacology

Faculty of Pharmaceutical Sciences, Sojo University, Japan

...................................................................................................

Comments and Suggestions for Authors

Undoubtedly, atopic dermatitis in children is an important topic in nutrition. The objective of the study was to determine the modulation of TH1/TH2/Treg immune response by EsA- and/or Esg-A- in splenocytes stimulated with ConA. The main concern is about the use of ConA as stimulus on splenocytes, mainly if the authors consider that this response could be representative of atopic dermatitis.

Of course, the present in vitro study could not be fully representative of atopic dermatitis model. However, as discussed in Discussion,「oral administration of esculeoside B, a solanocapsine-type glycoside and a major component in tomato juice, ameliorated experimental dermatitis in mice through decreases in IgE and ConA-mitogenic action, and a decline in IL-4 secretion [19-20]」, the present findings might underlie at least partly the mechanism on EsA-amelioration of experimental dermatitis.

I have some relevant observations about the model used by the authors:

Thank you for your indication.

  1. Do the authors consider the ConA stimulus appropriate to study atopic dermatitis?

Could you explain why you didn't use a more appropriate allergen?

We previously reported that oral administration of esculeoside B, a major component in tomato juice, ameliorated experimental dermatitis in mice through decreases in IgE and ConA-mitogenic action. In the next step, it is better for us to use a more appropriate allergen, such as ovalbumin or house dust mite.

  1. What is the reason for using only Balb/c females in the study?

Balb/c female mouse is preferred for immunology studies including T cell polarization, interleukin profiling and allergenicity. It has also been reported that female mice develop a more pronounced type of allergic inflammation than male mice. And our previous study related with the amelioration of atopic dermatitis by EsA also used Balb/c female mice.

  1. Were the mice exposed to any allergens or inducers of the atopic response?

The mice exposed to ConA for T cell mitogen, but not other inducers.

  1. The time for cytotoxic response is adequate. However, I think 24 hours is

inadequate for establishing a TH1 or TH2 immune response, especially if the mice were not previously in contact with an allergen. The authors need to explain why the response of naïve splenocytes stimulated with ConA would be characteristic of an individual with an atopic dermatitis profile?

Thank you for your advice. We also have investigated their responses for 48 or 72 hours, there are the similar effects with that for 24 hours.

It has been reported that ConA can stimulate mouse T-cell subsets giving rise to several functionally distinct T cell populations, including precursors to regulatory T cells [35]. Thus, the present findings might underlie at least partly the mechanism on EsA-amelioration of experimental dermatitis. These sentences are added in line 453-456, and one reference is added in line 578.

  1. Were cell death or apoptosis controls used in your flow cytometry studies or

splenocyte cultures? Please include them in the materials and methods sections and in the Results.

Considering the less cytotoxicity by EsA≦100 μM or by Esg-A≦10 μM from

Figure 2, we selected EsA≦100 μM or by Esg-A≦10 μM in Figure 6 of flow cytometry studies, and did not used cell death controls.

  1. Figures 3 and 5 could not be fully viewed in its pdf.

We corrected Figures 3 and 5.

  1. The authors mention that Esg-A can be excreted as androgen metabolites, could

the effects be due to an inhibition mediated by its analogy to glucocorticoids? have you detected levels of lipocortin or phospholipase A2?

Because the experiment on EsA metabolites were checked by using the urine of men

orally administered ripe cherry tomato fruit, we did not detect levels of lipocortin or phospholipase A2. It will be interesting to investigate such mechanism in the next step.

  1. In the Discussion section. The first part of the discussion is redundant with the

introduction and results section. The authors mention that EsA/Esg-A suppressed both Th2/Th1 cytokine production and intracellular gene expression levels, however, IFN-ϒ production was not analyzed (higher concentrations of Esg-A are missing in Figure 3).

We simplified the first part of the discussion.

IFN-ϒ production has been analyzed, but the data was not shown.

Figure 3 is corrected, I think you can see 30 μM of Esg-A. 

Dear Reviewer,

MDPI Nutrients Editorial Office

We address the following points, and have submitted a redline version of our manuscript.

We sincerely thank your pertinent advices in improving our manuscript.

Kind regards,

Jian-Rong Zhou

Department of Presymptomatic Medical Pharmacology

Faculty of Pharmaceutical Sciences, Sojo University, Japan

...................................................................................................

Comments and Suggestions for Authors

Undoubtedly, atopic dermatitis in children is an important topic in nutrition. The objective of the study was to determine the modulation of TH1/TH2/Treg immune response by EsA- and/or Esg-A- in splenocytes stimulated with ConA. The main concern is about the use of ConA as stimulus on splenocytes, mainly if the authors consider that this response could be representative of atopic dermatitis.

Of course, the present in vitro study could not be fully representative of atopic dermatitis model. However, as discussed in Discussion,「oral administration of esculeoside B, a solanocapsine-type glycoside and a major component in tomato juice, ameliorated experimental dermatitis in mice through decreases in IgE and ConA-mitogenic action, and a decline in IL-4 secretion [19-20]」, the present findings might underlie at least partly the mechanism on EsA-amelioration of experimental dermatitis.

I have some relevant observations about the model used by the authors:

Thank you for your indication.

  1. Do the authors consider the ConA stimulus appropriate to study atopic dermatitis?

Could you explain why you didn't use a more appropriate allergen?

We previously reported that oral administration of esculeoside B, a major component in tomato juice, ameliorated experimental dermatitis in mice through decreases in IgE and ConA-mitogenic action. In the next step, it is better for us to use a more appropriate allergen, such as ovalbumin or house dust mite.

  1. What is the reason for using only Balb/c females in the study?

Balb/c female mouse is preferred for immunology studies including T cell polarization, interleukin profiling and allergenicity. It has also been reported that female mice develop a more pronounced type of allergic inflammation than male mice. And our previous study related with the amelioration of atopic dermatitis by EsA also used Balb/c female mice.

  1. Were the mice exposed to any allergens or inducers of the atopic response?

The mice exposed to ConA for T cell mitogen, but not other inducers.

  1. The time for cytotoxic response is adequate. However, I think 24 hours is

inadequate for establishing a TH1 or TH2 immune response, especially if the mice were not previously in contact with an allergen. The authors need to explain why the response of naïve splenocytes stimulated with ConA would be characteristic of an individual with an atopic dermatitis profile?

Thank you for your advice. We also have investigated their responses for 48 or 72 hours, there are the similar effects with that for 24 hours.

It has been reported that ConA can stimulate mouse T-cell subsets giving rise to several functionally distinct T cell populations, including precursors to regulatory T cells [35]. Thus, the present findings might underlie at least partly the mechanism on EsA-amelioration of experimental dermatitis. These sentences are added in line 453-456, and one reference is added in line 578.

  1. Were cell death or apoptosis controls used in your flow cytometry studies or

splenocyte cultures? Please include them in the materials and methods sections and in the Results.

Considering the less cytotoxicity by EsA≦100 μM or by Esg-A≦10 μM from

Figure 2, we selected EsA≦100 μM or by Esg-A≦10 μM in Figure 6 of flow cytometry studies, and did not used cell death controls.

  1. Figures 3 and 5 could not be fully viewed in its pdf.

We corrected Figures 3 and 5.

  1. The authors mention that Esg-A can be excreted as androgen metabolites, could

the effects be due to an inhibition mediated by its analogy to glucocorticoids? have you detected levels of lipocortin or phospholipase A2?

Because the experiment on EsA metabolites were checked by using the urine of men

orally administered ripe cherry tomato fruit, we did not detect levels of lipocortin or phospholipase A2. It will be interesting to investigate such mechanism in the next step.

  1. In the Discussion section. The first part of the discussion is redundant with the

introduction and results section. The authors mention that EsA/Esg-A suppressed both Th2/Th1 cytokine production and intracellular gene expression levels, however, IFN-ϒ production was not analyzed (higher concentrations of Esg-A are missing in Figure 3).

We simplified the first part of the discussion.

IFN-ϒ production has been analyzed, but the data was not shown.

Figure 3 is corrected, I think you can see 30 μM of Esg-A. 

Dear Reviewer,

MDPI Nutrients Editorial Office

We address the following points, and have submitted a redline version of our manuscript.

We sincerely thank your pertinent advices in improving our manuscript.

Kind regards,

Jian-Rong Zhou

Department of Presymptomatic Medical Pharmacology

Faculty of Pharmaceutical Sciences, Sojo University, Japan

...................................................................................................

Comments and Suggestions for Authors

Undoubtedly, atopic dermatitis in children is an important topic in nutrition. The objective of the study was to determine the modulation of TH1/TH2/Treg immune response by EsA- and/or Esg-A- in splenocytes stimulated with ConA. The main concern is about the use of ConA as stimulus on splenocytes, mainly if the authors consider that this response could be representative of atopic dermatitis.

Of course, the present in vitro study could not be fully representative of atopic dermatitis model. However, as discussed in Discussion,「oral administration of esculeoside B, a solanocapsine-type glycoside and a major component in tomato juice, ameliorated experimental dermatitis in mice through decreases in IgE and ConA-mitogenic action, and a decline in IL-4 secretion [19-20]」, the present findings might underlie at least partly the mechanism on EsA-amelioration of experimental dermatitis.

I have some relevant observations about the model used by the authors:

Thank you for your indication.

  1. Do the authors consider the ConA stimulus appropriate to study atopic dermatitis?

Could you explain why you didn't use a more appropriate allergen?

We previously reported that oral administration of esculeoside B, a major component in tomato juice, ameliorated experimental dermatitis in mice through decreases in IgE and ConA-mitogenic action. In the next step, it is better for us to use a more appropriate allergen, such as ovalbumin or house dust mite.

  1. What is the reason for using only Balb/c females in the study?

Balb/c female mouse is preferred for immunology studies including T cell polarization, interleukin profiling and allergenicity. It has also been reported that female mice develop a more pronounced type of allergic inflammation than male mice. And our previous study related with the amelioration of atopic dermatitis by EsA also used Balb/c female mice.

  1. Were the mice exposed to any allergens or inducers of the atopic response?

The mice exposed to ConA for T cell mitogen, but not other inducers.

  1. The time for cytotoxic response is adequate. However, I think 24 hours is

inadequate for establishing a TH1 or TH2 immune response, especially if the mice were not previously in contact with an allergen. The authors need to explain why the response of naïve splenocytes stimulated with ConA would be characteristic of an individual with an atopic dermatitis profile?

Thank you for your advice. We also have investigated their responses for 48 or 72 hours, there are the similar effects with that for 24 hours.

It has been reported that ConA can stimulate mouse T-cell subsets giving rise to several functionally distinct T cell populations, including precursors to regulatory T cells [35]. Thus, the present findings might underlie at least partly the mechanism on EsA-amelioration of experimental dermatitis. These sentences are added in line 453-456, and one reference is added in line 578.

  1. Were cell death or apoptosis controls used in your flow cytometry studies or

splenocyte cultures? Please include them in the materials and methods sections and in the Results.

Considering the less cytotoxicity by EsA≦100 μM or by Esg-A≦10 μM from

Figure 2, we selected EsA≦100 μM or by Esg-A≦10 μM in Figure 6 of flow cytometry studies, and did not used cell death controls.

  1. Figures 3 and 5 could not be fully viewed in its pdf.

We corrected Figures 3 and 5.

  1. The authors mention that Esg-A can be excreted as androgen metabolites, could

the effects be due to an inhibition mediated by its analogy to glucocorticoids? have you detected levels of lipocortin or phospholipase A2?

Because the experiment on EsA metabolites were checked by using the urine of men

orally administered ripe cherry tomato fruit, we did not detect levels of lipocortin or phospholipase A2. It will be interesting to investigate such mechanism in the next step.

  1. In the Discussion section. The first part of the discussion is redundant with the

introduction and results section. The authors mention that EsA/Esg-A suppressed both Th2/Th1 cytokine production and intracellular gene expression levels, however, IFN-ϒ production was not analyzed (higher concentrations of Esg-A are missing in Figure 3).

We simplified the first part of the discussion.

IFN-ϒ production has been analyzed, but the data was not shown.

Figure 3 is corrected, I think you can see 30 μM of Esg-A. 

Dear Reviewer,

MDPI Nutrients Editorial Office

We address the following points, and have submitted a redline version of our manuscript.

We sincerely thank your pertinent advices in improving our manuscript.

Kind regards,

Jian-Rong Zhou

Department of Presymptomatic Medical Pharmacology

Faculty of Pharmaceutical Sciences, Sojo University, Japan

...................................................................................................

Comments and Suggestions for Authors

Undoubtedly, atopic dermatitis in children is an important topic in nutrition. The objective of the study was to determine the modulation of TH1/TH2/Treg immune response by EsA- and/or Esg-A- in splenocytes stimulated with ConA. The main concern is about the use of ConA as stimulus on splenocytes, mainly if the authors consider that this response could be representative of atopic dermatitis.

Of course, the present in vitro study could not be fully representative of atopic dermatitis model. However, as discussed in Discussion,「oral administration of esculeoside B, a solanocapsine-type glycoside and a major component in tomato juice, ameliorated experimental dermatitis in mice through decreases in IgE and ConA-mitogenic action, and a decline in IL-4 secretion [19-20]」, the present findings might underlie at least partly the mechanism on EsA-amelioration of experimental dermatitis.

I have some relevant observations about the model used by the authors:

Thank you for your indication.

  1. Do the authors consider the ConA stimulus appropriate to study atopic dermatitis?

Could you explain why you didn't use a more appropriate allergen?

We previously reported that oral administration of esculeoside B, a major component in tomato juice, ameliorated experimental dermatitis in mice through decreases in IgE and ConA-mitogenic action. In the next step, it is better for us to use a more appropriate allergen, such as ovalbumin or house dust mite.

  1. What is the reason for using only Balb/c females in the study?

Balb/c female mouse is preferred for immunology studies including T cell polarization, interleukin profiling and allergenicity. It has also been reported that female mice develop a more pronounced type of allergic inflammation than male mice. And our previous study related with the amelioration of atopic dermatitis by EsA also used Balb/c female mice.

  1. Were the mice exposed to any allergens or inducers of the atopic response?

The mice exposed to ConA for T cell mitogen, but not other inducers.

  1. The time for cytotoxic response is adequate. However, I think 24 hours is

inadequate for establishing a TH1 or TH2 immune response, especially if the mice were not previously in contact with an allergen. The authors need to explain why the response of naïve splenocytes stimulated with ConA would be characteristic of an individual with an atopic dermatitis profile?

Thank you for your advice. We also have investigated their responses for 48 or 72 hours, there are the similar effects with that for 24 hours.

It has been reported that ConA can stimulate mouse T-cell subsets giving rise to several functionally distinct T cell populations, including precursors to regulatory T cells [35]. Thus, the present findings might underlie at least partly the mechanism on EsA-amelioration of experimental dermatitis. These sentences are added in line 453-456, and one reference is added in line 578.

  1. Were cell death or apoptosis controls used in your flow cytometry studies or splenocyte cultures? Please include them in the materials and methods sections and in the Results.

Considering the less cytotoxicity by EsA≦100 μM or by Esg-A≦10 μM from

Figure 2, we selected EsA≦100 μM or by Esg-A≦10 μM in Figure 6 of flow cytometry studies, and did not used cell death controls.

  1. Figures 3 and 5 could not be fully viewed in its pdf.

We corrected Figures 3 and 5.

  1. The authors mention that Esg-A can be excreted as androgen metabolites, could

the effects be due to an inhibition mediated by its analogy to glucocorticoids? have you detected levels of lipocortin or phospholipase A2?

Because the experiment on EsA metabolites were checked by using the urine of men

orally administered ripe cherry tomato fruit, we did not detect levels of lipocortin or phospholipase A2. It will be interesting to investigate such mechanism in the next step.

  1. In the Discussion section. The first part of the discussion is redundant with the

introduction and results section. The authors mention that EsA/Esg-A suppressed both Th2/Th1 cytokine production and intracellular gene expression levels, however, IFN-ϒ production was not analyzed (higher concentrations of Esg-A are missing in Figure 3).

We simplified the first part of the discussion.

IFN-ϒ production has been analyzed, but the data was not shown.

Figure 3 is corrected, I think you can see 30 μM of Esg-A. 

Reviewer 2 Report

Major comments:
1. What I cannot understand is why all those in-vitro studies were performed on mouse cells. The end goal is always development of novel treatment regimens for human use -> thus, in this case, all those assays should have been performed on human PBMCs. 
2. Professional proofreading is needed. 
3. Please follow MinFlowCyt recommendations e.g. please provide clones for each mAb and cytometer configuration
4. Line 160: please provide details about fixation/permabilisation protocol.
5. Figure 4 qPCR results - is it 2^-dCt or 2^-ddCt?
6. Please clearly state limitations of your study within discussion

Minor comments:
1. "MeOH, 60% aq. MEOH and MeOH" - please write methanol instead. All abbreviations should be appropriately introduced and only necessary abbreviations should be used.
2. "The organic layer was evaporated in vacuo to afford a residue," - please rephrase
3. Please provide cat. numbers for ELISA kits
4. Line 126 should read RT-qPCR
5. Line 130 - RNEasy Mini Kit isolates TOTAL RNA
6. "fluorescence-labeled" - fluorochrome-labeled.
7. Since you are providing full names for substances like APC and CFSE, please do not ommitt FITC.

Author Response

Dear Reviewer,

MDPI Nutrients Editorial Office

We address the following points, and have submitted a redline version of our manuscript.

We sincerely thank your pertinent advices in improving our manuscript.

Kind regards,

Jian-Rong Zhou

Department of Presymptomatic Medical Pharmacology

Faculty of Pharmaceutical Sciences, Sojo University, Japan

...................................................................................................

Comments and Suggestions for Authors

Major comments:
1. What I cannot understand is why all those in-vitro studies were performed on mouse cells. The end goal is always development of novel treatment regimens for human use -> thus, in this case, all those assays should have been performed on human PBMCs. 

   Our previous study showed that EsA ameliorated experimental AD in mouse, so mouse cells were used for investigating the underlying mechanisms. It is important to study their effects on human PMBCs in the next step.

2. Professional proofreading is needed. 
   We thank Dr. Pernilla B for her language help, and try to improve our manuscript.

3. Please follow MinFlowCyt recommendations e.g. please provide clones for each mAb and cytometer configuration
   We provided clones for each mAb and also added their corresponding isotype control antibody for cytometer configuration in line 108, 110, 111, 163-165 and 168-169.

4. Line 160: please provide details about fixation/permabilisation protocol.
   The details were added in Line 165-167 and 169-170.

5. Figure 4 qPCR results - is it 2^-dCt or 2^-ddCt?
   It is 2-ΔΔCt.

6. Please clearly state limitations of your study within discussion

   The following sentences were added in Line 453-456 and Line 467-468.

   It has been reported that ConA can stimulate mouse T-cell subsets giving rise to several functionally distinct T cell populations, including precursors to regulatory T cells [35]. Thus, the present findings might underlie at least partly the mechanism on EsA-amelioration of experimental dermatitis.

   Next steps it is also important to study EsA/Esg-A effects on human PBMCs (peripheral blood mononuclear cell).

   One abbreviation and one reference are added in Line 473 and Line 578, respectively.

Minor comments:
1. "MeOH, 60% aq. MEOH and MeOH" - please write methanol instead. All abbreviations should be appropriately introduced and only necessary abbreviations should be used.

  MeOH is corrected as methanol in Line 69 and 73. AcOEt is corrected as ethyl acetate in Line 71.

2. "The organic layer was evaporated in vacuo to afford a residue," - please rephrase

  vacuum.

3. Please provide cat. numbers for ELISA kits

   Cat. Numbers were added in Line 125-126.

4. Line 126 should read RT-qPCR

  It is corrected in Line 128, 129 and 155.

5. Line 130 - RNEasy Mini Kit isolates TOTAL RNA

   It is corrected in Line 133.

6. "fluorescence-labeled" - fluorochrome-labeled.

   It is corrected in Line 163.

7. Since you are providing full names for substances like APC and CFSE, please do not ommitt FITC.

   The full names of FITC were added in Line 95 and Line 472.

Round 2

Reviewer 1 Report

  1. Regarding point 1, I did not find a justification for Con A use in the new version of the manuscript. Please include it, and if it is possible, have a reference that supports the fact that Con A can be used as a stimulus for this kind of immune response.
  2. I recommended including part of your answer in the corresponding section of the manuscript (2.2. Animals and splenocyte isolation).
  3. Maybe you misunderstood my question; the splenocytes were stimulated with Con A. However, ¿were the mice boosted before spleen dissection? I consider that the antigen used in the study was not appropriate. In most studies on atopic dermatitis, the model is cells extracted from patients with the condition. Murine models must be previously exposed to the antigen to induce this hypersensitivity.
  4. Please, check your pdf version before submitting your manuscript, now Figure 3C was partially missing.
  5. Have you quantified EsB in your preparations? This point is important since, according to what you have published, EsB has the same effect as the SpA reported in this manuscript (https://dx.doi.org/10.3390%2Fjpm12040579). Thus, even small concentrations of EsB could be part of the effect described in this work. Please include the evidence that your EsA preparation is EsB free in the manuscript.

Author Response

Dear Reviewer,

MDPI Nutrients Editorial Office

We address the following points, and have submitted a revised version of our manuscript.

We sincerely thank your pertinent advices in improving our manuscript.

Kind regards,

Jian-Rong Zhou

Department of Presymptomatic Medical Pharmacology

Faculty of Pharmaceutical Sciences, Sojo University, Japan

..................................................................................................

Comments and Suggestions for Authors

  1. Regarding point 1, I did not find a justification for Con A use in the new version of the manuscript. Please include it, and if it is possible, have a reference that supports the fact that Con A can be used as a stimulus for this kind of immune response.

“It has been also reported that ConA can stimulate mouse T-cell subsets giving rise to several functionally distinct T cell populations, including precursors to regulatory T cells [35]. Thus, the present findings might underlie at least partly the mechanism on EsA-amelioration of experimental dermatitis“. We added a justification for Con A use in the last version of the manuscript in line 450-452, and one reference was added in line 576-577.

  1. I recommended including part of your answer in the corresponding section of the manuscript (2.2. Animals and splenocyte isolation).

We added a justification for Con A use in the last version of the manuscript in line 450-452, and one reference was added in line 576-577.

  1. Maybe you misunderstood my question; the splenocytes were stimulated with Con A. However, ¿were the mice boosted before spleen dissection? I consider that the antigen used in the study was not appropriate. In most studies on atopic dermatitis, the model is cells extracted from patients with the condition. Murine models must be previously exposed to the antigen to induce this hypersensitivity.

Thank you for your advice. We will check such effects on other allergic diseases according to your suggestion in the next step.

  1. Please, check your pdf version before submitting your manuscript, now Figure 3C was partially missing.

It is corrected.

  1. Have you quantified EsB in your preparations? This point is important since, according to what you have published, EsB has the same effect as the SpA reported in this manuscript (https://dx.doi.org/10.3390%2Fjpm12040579). Thus, even small concentrations of EsB could be part of the effect described in this work. Please include the evidence that your EsA preparation is EsB free in the manuscript.

We have checked EsA in our preparation by thin-layer chromatography (TLC) method. Spirosolane-type EsA (line 29) and solanocapsine-type EsB (line 399) have different retention factor (Rf) values with the developing solvent CHCl3-methanol-H2O=6:4:1 as shown in the following figures. The TLC result of EsA is added to Figure 1 and Legend in line 75-77.

Reviewer 2 Report

Dear Authors, I am mostly satisfied with your responses. I have only two additional comments:

  1. What does "to afford a residue" mean?
  2. Please clearly state it in the limitations that those results may or may not be repeatable on human samples

Author Response

Dear Reviewer,

MDPI Nutrients Editorial Office

We address the following points, and have submitted a revised version of our manuscript.

We sincerely thank your pertinent advices in improving our manuscript.

Kind regards,

Jian-Rong Zhou

Department of Presymptomatic Medical Pharmacology

Faculty of Pharmaceutical Sciences, Sojo University, Japan

...................................................................................................

Comments and Suggestions for Authors

Dear Authors, I am mostly satisfied with your responses. I have only two additional comments:

  1. What does "to afford a residue" mean?

It means “to get a raw Esg-A”.

  1. Please clearly state it in the limitations that those results may or may not be repeatable on human samples.

The research from Dwyer et al [35] have suggested that humans as well as mice have distinctive subpopulations of suppressor cells triggered by Con A, thus it is hard to state that the present results may or may not be repeatable on human samples. However, our unpublished data shows that an oral administration of EsA ameliorated human atopic dermatitis, it is also important to study EsA/Esg-A effects on human in the next step.